# Use of Bedinvetmab (Librela^®^) for Canine Osteoarthritis in France, Germany, Italy, Spain, and the UK: Quantitative Analysis of Veterinarian Satisfaction and Real-World Treatment Patterns

**DOI:** 10.3390/ani14152231

**Published:** 2024-07-31

**Authors:** Edwina Gildea, Cyndy North, Kate Walker, Francis Adriaens, Benedict Duncan X. Lascelles

**Affiliations:** 1Zoetis UK Limited, 1st Floor, Birchwood Building, Springfield Drive, Leatherhead KT22 7LP, UK; 2Zoetis LLC, 10 Sylvan Way, Parsippany, NJ 07054, USA; 3Translational Research in Pain Program, Comparative Pain Research and Education Centre, Department of Clinical Sciences, College of Veterinary Medicine, North Carolina State University, Raleigh, NC 27606, USA

**Keywords:** bedinvetmab, Librela, canine osteoarthritis, real-world, pain, canine degenerative joint disease

## Abstract

**Simple Summary:**

Bedinvetmab (Librela^®^) is a new medicine for dogs who feel pain from osteoarthritis. The aim of this study was to understand how veterinarians select which dogs with osteoarthritis will get bedinvetmab, and whether the veterinarians are satisfied with this medicine. Overall, 1932 patient record forms (PRF) were collected from 375 veterinarians across five countries in Europe. Veterinarians provided 5–7 PRF each, representing an average dog that they gave bedinvetmab to. Veterinarians’ satisfaction level with bedinvetmab averaged 8.0 out of 10.0 across all countries. Fewer than 1% of veterinarians stopped prescribing bedinvetmab due to being unsatisfied. After starting bedinvetmab, the proportion of dogs needing more than one pharmacological medicine for osteoarthritis pain fell from 47% to 31% (*p* < 0.05). After starting bedinvetmab, the mean average total number of pharmacological medicines per dog reduced to 1.3 from 1.9 before starting (*p* < 0.05). Compliance and satisfaction levels appear high, and most dogs need fewer pharmacological medicines at one time to treat osteoarthritis pain after starting bedinvetmab.

**Abstract:**

Bedinvetmab (Librela^®^) represents a new class of canine osteoarthritis pain therapy. The aim of this study was to understand patient selection, usage behaviours, and satisfaction amongst veterinarians using bedinvetmab. Overall, 1932 patient record forms (PRF) were collected from 375 veterinarians across five countries in Europe. Veterinarians were asked to provide 5–7 PRF representing an average patient prescribed bedinvetmab. Veterinarian satisfaction with bedinvetmab usage averaged 8.0 out of 10.0 across all countries. Dissatisfaction as a reason for discontinuation was less than 1% for veterinarians. Veterinarians prescribed bedinvetmab broadly, across patient severity stages, weights, and ages. Adherence to monthly dosing per the product label was over 99%, and compliance with bedinvetmab treatment regimens was 85%. Following initiation of bedinvetmab, the proportion of patients requiring multiple pharmacological therapies for osteoarthritis pain fell from 47% to 31% (*p* < 0.05). After initiation of bedinvetmab, the mean total number of pharmacological therapies per patient across the population was 1.3, a reduction from 1.9 pre-treatment (*p* < 0.05). This investigation provides evidence on the benefit of bedinvetmab use post-launch in a broad population of dogs across the five most populous countries in western Europe. Compliance and satisfaction appear high and the use of other analgesic therapies to treat osteoarthritis pain is reduced in most cases following administration of bedinvetmab.

## 1. Introduction

Osteoarthritis (OA) is a chronic, progressive disease of synovial joints and the most common form of arthritis in both humans and dogs [1,2]. Clinically, OA is characterized by pain and stiffness of the joints, which can lead to a variety of clinical signs in affected dogs [3]. The clinical signs of OA pain can include a withdrawn demeanour, reduced sociability and play, altered posture, and mobility changes (e.g., stiffness and lameness), all of which compromise the dogs’ ability to perform the activities of daily living. These clinical signs can translate to a reduced quality of life (QoL) in affected dogs [3,4,5]. Recently, OA diagnosis has subsequently been reported in 38% of a sample of dogs who had previously shown one or more signs of related impairment [6].

OA management is typically multimodal, and can include the use of nutritional supplements and other interventions to promote a healthy body condition and increase or decrease physical activity (as appropriate), and pain relief through primarily pharmacological treatments [7]. To date, NSAIDs are a common first-line pharmacological option in OA pain management and have been shown to be beneficial in relieving joint pain, restoring mobility, and improving QoL [8,9].

Current NSAID treatment options are clearly effective but have some limitations [10,11,12,13,14]. NSAIDs are not always sufficiently effective on their own [15], and NSAIDs can also be associated with gastrointestinal, renal, and hepatic adverse events [16]. Compliance with daily oral medication may be difficult for pet owners [17]. The efficacy of NSAIDs improves with their use over time, and therefore challenges with compliance may prevent dogs from experiencing longer-term benefits [18,19,20]. These challenges with current treatments led to the search for new solutions to help veterinarians manage OA pain.

The development of OA pain is a complex process, mediated by many factors, including prostaglandins (e.g., PGE2) as well as nerve growth factor (NGF), a key signalling protein that is produced by injured tissues [21,22]. NGF is elevated in the osteoarthritic joints of multiple species [21,23]. Following injury, inflammatory cytokines and NGF are released by tissues of the joint. NGF binds to TrkA/p75NTR receptors found on peripheral nerves, immune cells, endothelial cells, synoviocytes, and chondrocytes. Binding to TrkA receptors on sensory nerves induces peripheral sensitization, heightens neurogenic inflammation when sensory nerves are stimulated, and significantly contributes to increased pain perception. NGF also binds to TrkA receptors located on immune cells to elicit the release of additional pro-inflammatory mediators, including NGF itself [21]. These inflammatory mediators lead to further peripheral sensitization involved in pain perception. Therefore, NGF is a prime target in this pathway for new drug development [21].

Bedinvetmab, the first medication in a new class of therapy targeting NGF, was approved in November, 2020 for veterinary use in the European Union and the United Kingdom (UK) for the alleviation of chronic OA pain in dogs [24,25]. Bedinvetmab is a first-in-class monoclonal antibody (mAb) designed for dogs that neutralizes NGF to alleviate OA pain [24]. Bedinvetmab sequesters NGF, thereby lowering the amount available to bind to the TrkA and p75NTR receptors, resulting in decreased signal transduction in cell types involved in pain [21,26]. Bedinvetmab is eliminated by the body similar to endogenous proteins, with minimal involvement of the liver or kidneys [27,28].

As bedinvetmab is the first in this new class of anti-NGF mAbs, real-world clinical data and usage patterns associated with this treatment are unknown. Therefore, this study sought to understand clinical real-world usage patterns within the first year of the introduction of bedinvetmab in the first markets to launch: France, Germany, Italy, Spain, and the UK.

## 2. Materials and Methods

A quantitative retrospective online survey was conducted by FMR Global Health to understand usage patterns of bedinvetmab for the alleviation of OA pain and veterinarian satisfaction with this therapeutic (see Figure 1) [29]. Each participating veterinarian was asked to submit a maximum of seven patient record forms (PRFs), each representing a unique patient treated with bedinvetmab.

The PRFs captured data on patient demographics (including disease severity), bedinvetmab usage (including frequency), and medications used prior to and after initiation of bedinvetmab on a per-patient basis. Veterinarian satisfaction on an individual patient basis and pet owner compliance with monthly dosing were also captured as part of these PRFs. Participating veterinarian identities were blinded to the study sponsor, and the study sponsor identity was never revealed to the participants in order to avoid potential bias.

Data pertaining to usage was retrospectively derived from February 2021 onwards, after bedinvetmab had become available to participating veterinarians. Collection of these data (that is, completion of record forms by veterinarians) occurred between October 2021 and November 2021.

No ethics approval was sought or required for this retrospective chart review market research study, as no animal or human research subjects were recruited, no intervention was made, and no identified or identifiable data were reported to the study sponsor or are being presented in this paper. Furthermore, as this study involved only retrospective data collection from veterinarians, no risk (physical, financial, or other) was anticipated to occur to any pet or pet owner.

### 2.1. Survey Respondents and Patient Record Selection

#### 2.1.1. Targeted Profile for Study Participants

General practitioner (GP) veterinarians included in the study were recruited from various sources to avoid the bias that can arise from a single panel sample. The sources used for recruitment included a veterinary panel, veterinary associations (e.g., Federation of Veterinarians of Europe, European Board of Veterinary Specialisation), local directories, clinic lists, and published literature. To mitigate bias, the sponsor did not have any involvement in the selection of respondents. The identity of participating veterinarians was also withheld from the study sponsor to maintain study blinding. Study participants were screened by telephone to ensure that a representative population was recruited via opportunistic sampling across all countries.

Veterinarians were then invited to participate via opportunistic sampling until a recruitment quota of 375 veterinarians was reached. Opportunistic sampling was employed as included veterinarians were required to have experiences of prescribing bedinvetmab in order to examine key trends in bedinvetmab usage. To mitigate bias, veterinarians were randomly recruited by phone to a target number of *n* = 75 per country and were required to meet specific criteria in terms of hours spent practicing weekly, years in practice, and number of unique canine patients seen monthly, to be eligible for participation in the study (see Table 1). The screening questionnaire was also used to ensure veterinarians had sufficient experience with bedinvetmab in order to participate in the study and to have possessed a sufficient supply of bedinvetmab to be able to provide consistent monthly dosing for their canine patients who started on the therapy. Therefore, participants were recruited according to criteria that ensured sufficient experience with both bedinvetmab and OA pain treatment more broadly.

#### 2.1.2. Timing of Bedinvetmab Prescription

To ensure that veterinarians recruited had longer-term experience using bedinvetmab, and to be able to measure pet owner compliance, at least 50% of study respondents were required to have begun prescribing bedinvetmab between February and April 2021. Veterinarians who began prescribing bedinvetmab between May and September 2021 comprised the remainder of respondents.

#### 2.1.3. Patient Record Selection

Participating veterinarians were asked to provide 5–7 PRFs for patients they believed represented typical cases treated with bedinvetmab in their practice (see Table 2).

As this was a retrospective study, veterinarians had the flexibility to select cases where they reduced, continued, or added treatments alongside bedinvetmab administration. Veterinarians completed the forms for representative dogs with OA pain that they had seen between February and November 2021 at one time, retrospectively. This approach ensured the data collected reflected the veterinarians’ experience from the first to the most recent bedinvetmab dose received by the patient during this time.

### 2.2. Patient Record Forms

Participating veterinarians were asked to complete PRFs that collected information on patient demographics, bedinvetmab administration and dosage, and veterinarian bedinvetmab satisfaction on a per-patient basis. Veterinarian satisfaction was assessed using a 10-point scale, whereby veterinarians were asked to rate their level of satisfaction from 1 (not at all satisfied) to 10.0 (fully satisfied). Information on pharmaceutical medications received for OA prior to initiation of bedinvetmab (if any), as well as changes in medications occurring concurrently with bedinvetmab treatment, were also collected. The PRFs also collected patient age, sex, weight, and OA case severity for each patient when first diagnosed with OA. Details relating to the method of staging dogs with OA, key signs leading to suspicion or confirmation of OA diagnosis by the veterinarian, and practice-determined requirements following diagnosis (e.g., need for regular blood work) were also collected. As these are real world data, veterinarians may have categorized a dog as suspected to have OA based on clinical signs but lacking radiographic confirmation.

Of note, determination of mild/moderate/severe disease status involved assigning each case to one of three distinct descriptions of hypothetical OA cases (outlining signs and behaviours associated with each severity level), provided as part of each PRF; veterinarians may or may not have also used a formal staging tool as part of their practice to facilitate this determination.

PRFs were provided in the participating veterinarian’s native language. A copy of a PRF (in English) is provided as Appendix A.

#### 2.2.1. Data Collection, Security, Confidentiality, and Quality Assurance

All patient records were collected electronically via a unique survey link while each veterinarian was in their clinic. The extent of data collection was designed to ensure a level of security appropriate to the potential risks for this study. Security and confidentiality measures included anonymisation, pseudonymisation (e.g., tokenisation), and encryption of personal data. All patient records were anonymised and not disclosed to the study sponsor or anyone outside the market research team. Personally identifiable information of veterinarians was also not shared with the study sponsor or anyone outside of the market research team. All data were processed and maintained in line with European Union General Data Protection Regulation requirements to ensure data security and confidentiality measures were adhered to within this study.

#### 2.2.2. Data Processing and Quality Assurance

When data collection was completed, the database was exported into both Microsoft Office Excel (version 2406) and SPSS (version 26) formats. Data cleaning and processing was then conducted, with additional checks on the quality and accuracy of the data. In case of contradictory or missing data, respondents were re-contacted for clarification by telephone. Outliers were identified statistically in the database, and any numerical responses that fell outside of the 95% confidence interval range were considered outliers and removed from the analysis. Typographical errors and logical errors were also identified, verified, and removed where necessary. Data cleaning and assessment steps were applied to ensure accurate data analysis.

Phone calls were used to follow up on any survey responses, if required, after data analysis. During fieldwork, quality checks were applied on a frequent basis to ensure the results were coherent and realistic.

### 2.3. Data Analysis

Unadjusted descriptive analyses were conducted to describe the characteristics of participating veterinarians and patients. Categorical variables are presented as numbers and percentages, while continuous variables are presented as means and standard deviations. Proportions of veterinarians are described in relation to the total number of participating veterinarians, while proportions of patients are described in relation to the total number of PRFs. Statistical significance of changes in medication prescription were assessed using paired *t*-tests at a 95% confidence interval (equivalent to *p* < 0.05) within SPSS Statistics software.

## 3. Results

### 3.1. 1932 PRFs Were Collected from 375 Veterinarians in the Five Countries

A total of 375 veterinarians took part in the study, each providing 5 to 7 PRFs. Overall, 1,932 PRFs were collected across all 5 countries (France, Germany, Italy, Spain, and the UK). Respondent veterinarians were from a mixture of clinic types (i.e., private, or corporate) and comprised of various profiles and decision-making roles (see Table 3). Disease and prescribing data were obtained through PRFs, and analyses were conducted based on data extracted from PRFs across countries (see Table 4).

### 3.2. Over Half of Patients Received Other Therapies before Initiation of Bedinvetmab

A total of 1199 (62%) patients had received previous treatment before starting bedinvetmab (see Table 5). Prior to initiation of bedinvetmab, the mean total number of drugs per-patient across the entire population was 1.9 and the mean total number of drugs per-patient was 1.5, 1.8, and 2.2 for patients with mild, moderate, or severe OA pain, respectively. Type of therapy received before initiation of bedinvetmab varied across disease stages (see Table 5).

#### 3.2.1. NSAIDs Were the Most Frequently Prescribed Medication Prior to Initiation of Bedinvetmab

Of the treatments used before initiation of bedinvetmab, the most common oral therapies were NSAIDs: Metacam^®^ (46.7%), Previcox^®^ (32.4%), Galliprant^®^ (26.5%), and Onsior^®^ (14.7%). Nutraceuticals were used less frequently (see Table 5).

The mean duration of treatment was 56 days and 30 days for oral and injectable NSAID formulations, respectively. Metacam^®^ (the most used oral therapy overall; 46.7%) was given over a mean duration of 58 days.

#### 3.2.2. Non-NSAID Therapies Were Also Prescribed before Initiation of Bedinvetmab (Particularly Physical Therapy)

The most used non-NSAID oral therapies, gabapentin (14.2% of all patients) and tramadol (8.5% of all patients), were prescribed for an average of 62 and 54 days, respectively. Cosequin^®^ (27.7% of all patients), the most used nutraceutical, was prescribed for an average of 85 days. Other therapies used prior to bedinvetmab initiation included hydrotherapy (46.2% received, for a mean duration of 37 days), laser treatment (28.0% received, for a mean duration of 26 days), and cannabidiol oil (10.8% received, for a mean duration of 39 days) among others (see Table 5).

### 3.3. Bedinvetmab Was Initiated as First-Line Therapy or Following Lack of Response to Previous Therapy

As per the label, almost all patients received bedinvetmab according to a monthly dosing schedule (99%), with only 12 (<1%) patients receiving bimonthly dosing. Dogs began therapy at different times in 2021 and the mean number of dose administrations across all countries was 4 (range: 1–10). Dosing intervals and dosing by weight were consistent with the product label.

Reasons for initiation of bedinvetmab treatment regimens were recorded (including free-text responses). The three most frequently given reasons for initiation of bedinvetmab were the patient being unable to reach a complete reduction in pain following inadequate response to previous therapies (40%), initiation of osteoarthritic therapy for the first time (40%), or to improve treatment compliance (28%).

#### 3.3.1. Bedinvetmab Was Used across Age, Weight, and OA Pain Severity Levels

Most patients treated with bedinvetmab were aged 7–12 years (63.7%). Other age categories represented were 2–6 years (8.9%), 13–17 years (23.8%), and 18–21 (3.6%) years (see Table 4). Within this patient population, bedinvetmab was prescribed to patients across all weight categories (<10 kg, 11–20 kg, 21–30 kg, 31–40 kg, 41–50 kg, and >50 kg). The 11–20 kg (23% of patients) and 21–30 kg (27% of patients) categories were represented the most within the PRFs received (see Table 4). Patients with mild, moderate, and severe OA pain comprised approximately one third of the study population each (27–37%; see Table 4).

#### 3.3.2. Most Patients Were Compliant with Their Treatment Regimen

Overall, 85% of pet owners complied with the bedinvetmab monthly treatment regimen that was prescribed (see Figure 2). Non-compliance was due to scheduling difficulties and interference with the daily life of the pet owner (9%), veterinarians recommending less frequent dosing (3%), cost issues (3%), improvement in presentation of clinical signs without therapy (2%), or pet owners choosing to discontinue therapy (1%).

### 3.4. Overall Veterinarian-Reported Satisfaction with Bedinvetmab (on a Ten-Point Scale) Appeared High in This Study and Increased at Later Doses

A rating of 8.0 out of 10.0 was selected by the greatest proportion of veterinarians (42.1%), followed by a rating of 9.0 (27.9%) and 7.0 (20.2%) across all countries. A total of 2% of veterinarians across all countries selected a satisfaction rating ≤5.0 (see Table 6). The difference in veterinarian satisfaction levels between patients who received previous treatment before bedinvetmab initiation and N patients who did not receive previous treatment was similar, with a mean difference of 0.2 (8.1 vs. 7.9 out of 10.0).

Comparing veterinarian satisfaction by number of bedinvetmab doses, mean satisfaction levels stayed the same or increased numerically at each additional dose. Mean satisfaction was 7.9 out of 10.0 across cases where one to four bedinvetmab doses were prescribed (*n* = 1280), and 8.2 out of 10.0 across cases where five or more bedinvetmab doses were prescribed (*n* = 625).

The mean satisfaction rating from veterinarians for bedinvetmab use was 7.9 out of 10.0 for use in mild patients, 8.0 out of 10.0 for use in moderate patients, and 8.1 out of 10.0 for use in severe patients. For bedinvetmab use in mild patients, a satisfaction rating of 8.0 out of 10.0 was the most common rating, selected by 40.1% of veterinarians, followed by 9.0 (26.9%) and 7.0 (21.4%). For bedinvetmab use in moderate patients, a satisfaction rating of 8.0 out of 10.0 was the most common rating, selected by 42.0% of veterinarians, followed by 9.0 (27.9%) and 7.0 (19.8%) out of 10.0. For bedinvetmab use in severe patients, a satisfaction rating of 8.0 out of 10.0 was the most common rating, selected by 45.0% of veterinarians, followed by 9.0 (29.2%) and 7.0 (19.1%) out of 10.0.

Veterinarian satisfaction with bedinvetmab was consistent across all weight categories (7.8–8.1 out of 10.0) with the lowest mean observed when used in patients in the ≤10 kg category (7.8) and highest means observed in the 41–50 kg categories and >50 kg (8.1 out of 10.0 for both groups). Veterinarian satisfaction was 0.3 times greater for bedinvetmab treatment of dogs in the heaviest weight category (>50 kg [8.1]) than for the lightest weight category (≤10 kg (7.8)).

Veterinarian satisfaction with bedinvetmab was also numerically consistent across patient age categories, with a mean rating of ≥7.0 out of 10.0 for all age categories (2–21 years). Veterinarian satisfaction with bedinvetmab also increased with patient age, with a mean rating of 7.0 out of 10.0 for dogs aged 2 years, and a mean rating 1.5 points higher for dogs aged 21 years (8.5 out of 10.0).

A total of 10 PRFs reported veterinarians expressing a satisfaction rating of <5, which were related to a lack of reduction in the patient’s pain, a lack of reduction in disease progression, or a negative response to therapy (e.g., aggression), and a need for additional therapies. These data demonstrate that veterinarian satisfaction level was consistent across all patient types receiving bedinvetmab, regardless of differences in weight, age, or OA severity.

### 3.5. Most Patients Did Not Require Other Therapies Following Initiation of Bedinvetmab; a Minority Added Other Therapies, or Continued Previous Therapies

Of the total 1932 patients, 75.0% received monotherapy for OA pain with bedinvetmab compared to 46.0% (551/1199) receiving a monotherapy prior to its availability (see Table 7). This percentage is inclusive of both dogs who had been treated for OA pain prior to bedinvetmab availability as well as newly treated and/or diagnosed patients. Of the 483 patients who did receive additional therapy, some dogs continued (189 [9.8%]), discontinued (22 [1.1%]), or added (280 [14.5%]) treatments concurrently (see Table 8, Table 9 and Table 10). Overall, following initiation of bedinvetmab, the proportion of patients requiring multiple pharmacological therapies for OA pain fell from 47% to 31% (*p* < 0.05 by paired *t*-test; see Table 11).

Of the 1199 patients who had received previous treatment of any type, the most common treatment continued was physical therapy (703 [58.6%]), followed by nutraceuticals (137 [11.4%]). Prior to the initiation of bedinvetmab, 1006 (83.9%) patients received oral drugs and 229 (19.1%) patients received injectable drugs to control OA pain. Less than 10% of the total population of patients who previously received oral or injectable treatments continued the same therapy they were receiving previously, following initiation of bedinvetmab (see Table 7). Focusing on NSAIDs, 17 patients discontinued, 36 patients continued, and 29 patients added these therapies following initiation of bedinvetmab. Non-NSAID oral medication use across countries varied, with tramadol use (brand not specified) being most frequently continued in patients in all countries (7% in France, 7% in Italy, 24% in Germany, 9% in Spain, and 18% in the UK).

After initiation of bedinvetmab, the mean total number of pharmacological medications per patient across the population was 1.3, a reduction from 1.9 pre-treatments (*p* < 0.05 by paired *t*-test; see Table 12). The mean total number of medications per-patient following initiation of treatment with bedinvetmab was 1.3, 1.3, and 1.4 for patients with mild, moderate, or severe OA pain, respectively (see Table 12).

## 4. Discussion

To the authors’ knowledge, the present study is the first to characterize usage patterns and outcomes relating to bedinvetmab in cases of canine OA pain since this treatment option became available in Europe.

This study shows that veterinarians who are prescribing bedinvetmab are prescribing bedinvetmab as a therapy for a wide range of dogs across a variety of characteristics, including different weights, ages, and disease severities. Following the launch of bedinvetmab in the countries of interest, common reasons for treatment initiation were to more effectively reduce pain, manage OA clinical signs in treatment-naive patients, and improve compliance in previously non-compliant patients and pet owners. Overall, veterinarian satisfaction with bedinvetmab was rated 8–10 in 73.5% of PRFs, while dissatisfaction (rating <5) was reported in fewer than 1%.

Bedinvetmab was used as a first line therapy in 40% of the cases, which reflects recommendations in the guidelines put forth by both the American Animal Hospital Association and the World Small Animal Veterinary Association in 2022 [9,30]. The majority of patients adhered to the monthly dosing outlined on the label for treatment with bedinvetmab (99%), greatly increasing the number of days dogs were on/receiving treatment compared with oral NSAIDs (average of 56 days). Overall, 85% of patients complied with all doses outlined in the per label monthly dosing regimen across Europe, with just 15% reported as having skipped doses of bedinvetmab [24]. Non-compliance (i.e., missing doses) was largely driven by factors such as scheduling difficulties, veterinarians recommending less frequent dosing, an improvement in the presentation of clinical signs, and interference with the daily life of the owner, pet owners choosing to discontinue therapy, or costs associated with OA treatment.

Following the initiation of bedinvetmab, only 65 patients (3.4%) continued with or added NSAID medication to their treatment regimen, highlighting that treatment with bedinvetmab may reduce the need for use of multiple medications to manage OA pain, thus serving to alleviate the burden of care and potential costs for the pet owner. Multi-modal management of OA is still recommended.

Prior to and with the introduction of bedinvetmab, the mean total number of drugs per-patient across the entire population dropped from 1.9 to 1.3 and the mean total number of drugs per-patient dropped from 1.5, 1.8, and 2.2 for patients with mild, moderate, or severe OA pain, respectively, to 1.3, 1.3, and 1.4, respectively. In addition, the number of dogs reported to have their pain managed with only one drug went from 46% to 75%, highlighting the need for additional medications prior to the initiation of Librela.

Currently, treatment of OA pain often includes polypharmacy, which is supported by the large variation of case characteristics and treatment experience in this study [31]. Pet owners may purchase over-the-counter medication and/or supplements to aid their dog’s wellbeing in addition to prescribed medication [32]. These behaviours were observed in qualitative interviews conducted with 40 UK pet owners, which demonstrated pet owner level of worry increased over time regarding their dog’s condition. This was associated with an increased need to physically assist their dog, as well as purchasing adjunctive therapies in addition to those already prescribed by their veterinarian [32].

Reduction in the number of medications that must be concurrently administered has been shown to reduce pet owner burden and improve compliance in other diseases [33], and the same is likely true in the management of chronic osteoarthritis. The reduction in the average number of therapies received following bedinvetmab treatment, along with the higher average duration of therapy with bedinvetmab observed in the current survey, suggests bedinvetmab is associated with a higher level of compliance than other pharmacological medications.

While this study may be useful to understand prescribing habits, clinical usage, and perceptions of efficacy, some methodological limitations and potential biases should be considered. This study followed an observational case series approach, where participating veterinarians shared data on representative cases treated with bedinvetmab. This study type is inherently less able to prove associations between interventions and outcomes when compared to more formal randomized designs. However, these data collected on key trends in prescribing of bedinvetmab are of real-world interest to inform veterinarian decision making in the future. A clear potential bias of this study is that, to be included, veterinarians were required to be ‘bedinvetmab users’, thus potentially biasing the results towards veterinarians who were satisfied with bedinvetmab. Conversely, this research may also have provided an opportunity for any veterinarians who were particularly dissatisfied with bedinvetmab to report their concerns; however, very few such results were collected.

This study collated insights shared by veterinarians who were willing to participate in this primary research via opportunistic sampling. As such, there is the possibility that the findings may be influenced by “volunteer” or “self-selection” bias, where individuals who volunteer for a study may share certain characteristics that are less generalisable to the eligible population. For example, the veterinarians who volunteered to take part may represent a subset of “early adopters” who are willing to utilize novel therapies in dogs with clinical characteristics or disease signs. Therefore, results may not be representative of all prescribing and usage trends across France, Germany, Italy, Spain, and the UK.

Additionally, opportunistically selected veterinarians were asked to provide a set of “the most average cases” of dogs receiving bedinvetmab; specifically, veterinarians were asked to consider 5–7 “average” patients. This may have also influenced the outcomes of this study by introducing slight availability and selection biases, as patients who are responding favourably may be more likely to return and continue on bedinvetmab. The request for the selection of “average cases” may have also introduced a level of subjectivity as veterinarians were asked to select representative cases at their own discretion. However, veterinarians were not incentivised to fulfil any prespecified profile, and selected cases were based on their clinical judgement. Additionally, veterinarians had the flexibility to select cases where they reduced, continued, or added treatments alongside bedinvetmab administration, which may have minimized the risk of systematic bias in this area.

Additionally, the study explored usage trends in patients currently receiving bedinvetmab, which may have excluded any patients for whom bedinvetmab was discontinued, and therefore findings may not be generalisable to bedinvetmab use in patients that have previously discontinued this therapy. However, the study does provide valuable insight into how bedinvetmab is prescribed when it is used in a practice setting, information which is likely to be of interest to those considering how this treatment option might be incorporated into their practices.

This research also relied on retrospective self-reporting by veterinarians that had utilized bedinvetmab following the launch of the therapy, who may have been influenced by social desirability or the urge to retrospectively reflect on certain patients who had been treated. As such, this retrospective study may have been associated with recall bias.

Veterinarians were asked to complete PRFs in relation to a particular therapy, bedinvetmab. While participants were blinded to the study sponsor, there is a high likelihood that the study sponsor may have been inferred by the respondents, based on the therapy of interest.

Finally, this research explored veterinarians’ clinical usage patterns within the first year of the introduction of bedinvetmab across France, Germany, Italy, Spain, and the UK; therefore, further research will be needed to explore the long-term clinical usage beyond this stage.

## 5. Conclusions

Following the review of 1,932 PRFs from 375 veterinarians across Europe who had been using bedinvetmab since its launch, these results suggest a high degree of satisfaction with bedinvetmab (expressed on a ten-point scale) as a therapy for dogs with OA pain. Bedinvetmab was initiated due to inadequate compliance with prior treatments in many cases, and high compliance with the recommended regimen of bedinvetmab was observed across Europe. Veterinarian satisfaction with bedinvetmab was observed in a large proportion of cases across countries, disease severity levels, and other patient demographics. Following bedinvetmab initiation, there was a significant reduction in the additional number of pharmacological medications used. This real-world insight will prove useful for practitioners considering the use of this therapeutic option.

## Figures and Tables

**Figure 1 animals-14-02231-f001:**
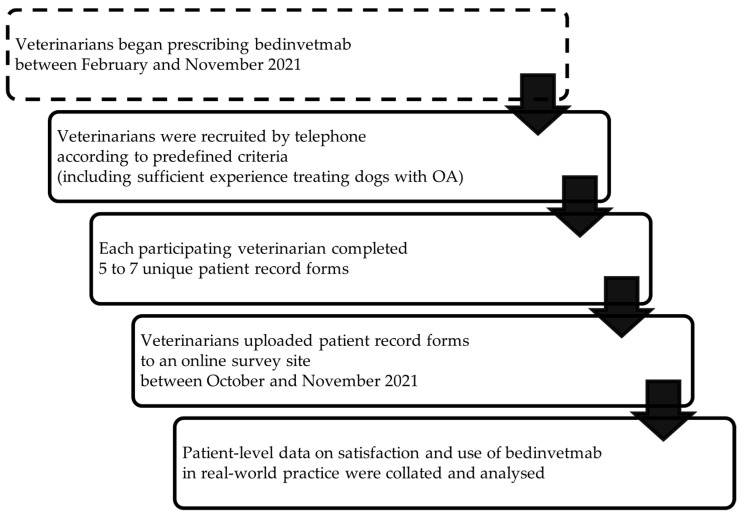
Flow of study activities represented in chronological order: prescribing and recruitment, record form completion and submission, and patient-level data analysis.

**Figure 2 animals-14-02231-f002:**
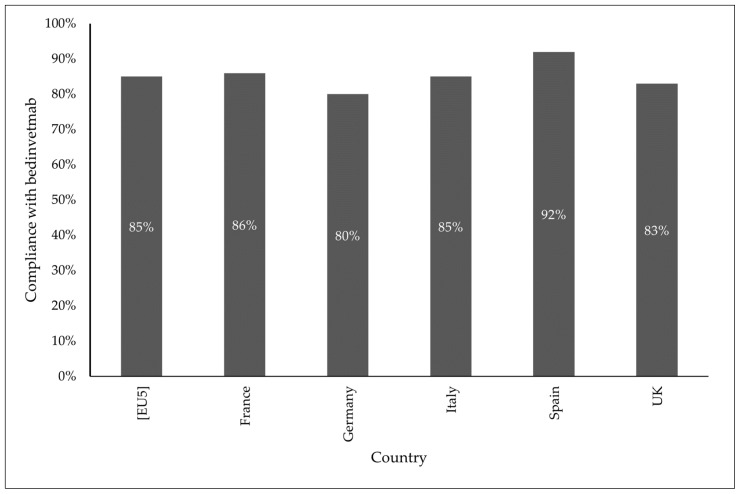
Compliance with bedinvetmab therapy across countries. Compliance is defined as the total percentage of patients who abided with product label dosages. From left to right the compliance percentages included are the total (EU5) percentage and individual percentages reported in each country (France, Germany, Italy, Spain, UK).

**Table 1 animals-14-02231-t001:** Screening criteria applied for veterinarian respondents.

GP Veterinarian Screening Criteria
Had to be a veterinary practice owner, partner, salaried veterinarian, veterinary clinic associate, or employee.
Must be licensed and have work experience in a veterinary practice for at least 3 years and up to 35 years.
The majority of their time working had to be in private practice (≥20 h per week).
At least 80% of their time must be dedicated to the treatment and care of dogs and cats.
They must treat a minimum of 40 dogs per month with at least ten patients per month being treated for OA.

OA: osteoarthritis.

**Table 2 animals-14-02231-t002:** Screening criteria applied for patients ^1^.

Patient Criteria (for Submitted Record Forms)
Must have a confirmed diagnosis or suspicion of OA (including pain associated with OA).
Must have a current prescription for bedinvetmab.
Patient must be at least 12 months of age.
Patient must not be currently pregnant, or lactating.

^1^ Veterinarians who had more than 5–7 patients in their practice who met all other criteria were requested to choose 5–7 patient forms they deemed average cases and describe why.

**Table 3 animals-14-02231-t003:** Veterinarian participant demographics.

Subgroup	France	Germany	Italy	Spain	UK	Total
Population size	75 (20%)	75 (20%)	75 (20%)	75 (20%)	75 (20%)	375 (100%)
Veterinarian type						
Salaried veterinarian (%) ^1^	42.7%	37.3%	0%	0%	34.7%	22.9%
Veterinarian associate or employee of veterinarian clinic (%) ^2^	38.7%	44%	62.7%	73.3%	40%	51.7%
Veterinarian—practice owner or partner (%) ^3^	18.7%	18.7%	37.3%	26.7%	25.3%	25.3%
Hours per week						
20–30	41 (54.7%)	49.3 (65.3%)	39 (52%)	43 (57.3%)	41 (54.7%)	213 (56.8%)
30+	34 (45.3%)	26 (34.7%)	36 (48%)	32 (42.7%)	34 (45.3%)	162 (43.2%)
Clinic setting						
Privately owned	57 (76%)	56 (74.7%)	75 (100%)	75 (100%)	49 (65.3%)	312 (83.2%)
Corporately owned	18 (24%)	19 (25.3%)	0 (0%)	0 (0%)	26 (34.7%)	63 (16.8%)
Proportion of veterinarians using products						
Bedinvetmab (initiated February to April 2021)	55%	68%	55%	68%	61%	61%
Bedinvetmab (initiated May 2021 or later)	45%	32%	45%	32%	39%	39%
Metacam^®^	85%	85%	81%	93%	93%	88%
Previcox^®^	89%	84%	79%	77%	92%	84%
Onsior^®^	60%	57%	79%	57%	53%	61%
Rimadyl^®^	61%	56%	77%	67%	53%	63%
Galliprant^®^	73%	76%	73%	84%	69%	75%
“Generic NSAID”	13%	20%	43%	35%	53%	33%
Other	51%	63%	67%	60%	76%	63%
Average number of dogs receiving treatment per clinic						
Bedinvetmab (initiated February to April 2021)	2	2	1	1	2	1
Bedinvetmab (initiated May 2021 or later)	13	14	12	12	10	12
Metacam^®^	69	67	62	71	94	72
Previcox^®^	50	47	45	42	59	49
Onsior^®^	21	26	24	18	18	22
Rimadyl^®^	18	16	17	14	13	16
Galliprant^®^	31	34	26	27	34	30
“Generic NSAID”	2	6	4	7	10	6
Other	12	21	16	14	28	18

NSAID: non-steroidal anti-inflammatory drug. ^1^ Salaried vet: a veterinarian on payroll for one practice. ^2^ Veterinary associate or employee of veterinary clinic: veterinary consultancy workers who are not listed to one practice, and receptionists. ^3^ Veterinary practice owner or partner: owners of one or more clinics and business partners of owners.

**Table 4 animals-14-02231-t004:** Patient record form demographics (including respondent country and patient characteristics).

Subgroup	France	Germany	Italy	Spain	UK	Total
Total number, n (% of total population)	381 (19.4%)	384 (19.7%)	390 (20.4%)	387 (20.1%)	390 (20.4%)	1932 (100%)
Patient characteristics for each country ^1^						
Male	244 (64%)	238 (62%)	278 (71%)	282 (73%)	253 (64%)	1295 (67%)
Female	137 (36%)	146 (38%)	112 (29%)	105 (27%)	137 (35%)	637 (33%)
Mean age (years)	11.5 (3.3–20.7)	11.4 (2.5–20.8)	10.1 (2.7–17)	10.6 (2.5–18.3)	10.8 (2.3–20.1)	11 (2.3–20.8)
≥2 to <7 years	28 (7.3%)	32 (8.3%)	40 (10.3%)	23 (5.9%)	49 (12.6%)	172 (8.9%)
≥7 to <13 years	222 (58.3%)	228 (59.4%)	277 (71.0%)	279 (72.1%)	225 (57.7%)	1,231(63.7%)
≥13 to <18 years	104 (27.3%)	99 (25.8%)	73 (18.8%)	79 (20.4%)	104 (26.7%)	459 (23.8%)
≥18 to <22 years	27 (7.1%)	25 (6.5%)	0 (0%)	6 (1.6%)	12 (3.1%)	70 (3.6%)
Mean weight (kg)	31.9 (6.0–72.0)	32.4 (6.0–63.0)	28.9 (6.0–72.0)	31.0 (6.0–65.0)	28.2 (8.0–64.0)	30.4 (6.0–72.0)
≤10 kg	13 (3%)	8 (2%)	26 (7%)	15 (4%)	21 (5%)	83 (4%)
11–20 kg	72 (19%)	78 (20%)	101 (26%)	81 (21%)	107 (27%)	439 (23%)
21–30 kg	105 (28%)	97 (25%)	101 (26%)	105 (27%)	122 (31%)	530 (27%)
31–40 kg	87 (23%)	89 (23%)	89 (23%)	95 (25%)	66 (17%)	426 (22%)
41–50 kg	60 (16%)	68 (18%)	44 (11%)	68 (18%)	51 (13%)	291 (15%)
>50 kg	44 (12%)	44 (11%)	29 (7%)	23 (6%)	23 (6%)	163 (8%)
No comorbidities	30 (8%)	13 (3%)	102 (26%)	38 (10%)	54 (14%)	237 (12%)
Comorbidities	351 (92%)	371 (97%)	288 (84%)	349 (90%)	336 (86%)	1695 (88%)
Oral infection (tartar and gingivitis)	94 (25%)	144 (38%)	94 (24%)	142 (37%)	137 (35%)	611 (32%)
Ear infection	56 (15%)	177 (46%)	27 (7%)	164 (42%)	54 (14%)	478 (25%)
Itchy skin/skin infections	105 (28%)	196 (51%)	62 (16%)	203 (52%)	124 (32%)	690 (36%)
Urinary problems	149 (39%)	210 (55%)	98 (25%)	208 (54%)	162 (42%)	827 (43%)
Obesity	50 (13%)	63 (16%)	17 (4%)	67 (17%)	42 (11%)	239 (12%)
Diabetes	108 (28%)	86 (22%)	36 (9%)	68 (18%)	45 (12%)	343 (18%)
Cardiac disease	84 (22%)	86 (22%)	55 (14%)	40 (10%)	60 (15%)	325 (17%)
Renal impairment	80 (21%)	60 (16%)	65 (17%)	81 (21%)	50 (13%)	336 (17%)
Other	67 (18%)	40 (10%)	151 (39%)	84 (22%)	86 (22%)	428 (22%)
Disease characteristics ^1^						
OA suspected by veterinarian	198 (52%)	194 (51%)	169 (43%)	181 (47%)	136 (35%)	878 (45%)
OA diagnosed by veterinarian	183 (48%)	190 (49%)	221 (57%)	206 (53%)	254 (65%)	1054 (55%)
Disease severity ^1^						
Mild (early stage) OA	127 (33%)	141 (37%)	162 (42%)	127 (33%)	139 (36%)	696 (36%)
Moderate (mid-stage) OA	150 (39%)	133 (35%)	135 (35%)	144 (37%)	150 (38%)	712 (37%)
Severe (late stage) OA	104 (27%)	110 (29%)	93 (24%)	116 (30%)	101 (26%)	524 (27%)
Diagnostic measures ^1^						
Staging tool used (No)	320 (84%)	330 (86%)	313 (80%)	293 (76%)	293 (75%)	1549 (80%)
Staging tool used (Yes)	61 (16%)	54 (14%)	77 (20%)	94 (24%)	97 (25%)	383 (20%)
COAST staging tool	0 (0%)	0 (0%)	0 (0%)	0 (0%)	5 (5%)	5 (1%)
CT scan	23 (38%)	18 (33%)	35 (45%)	18 (19%)	33 (34%)	127 (33%)
Joint fluid analysis	0 (0%)	1 (2%)	4 (5%)	13 (14%)	4 (4%)	22 (6%)
MRI	0 (0%)	6 (11%)	1 (1%)	5 (5%)	10 (10%)	22 (6%)
X-ray	46 (75%)	35 (65%)	51 (66%)	71 (76%)	65 (67%)	268 (70%)
Management ^1^						
Mean number of times seen for OA	6 (1–14)	7 (2–16)	6 (1–24)	7 (1–15)	8 (1–25)	7 (1–25)
Blood work (every 6 months)	113 (30%)	104 (27%)	15 (4%)	48 (12%)	43 (11%)	323 (17%)
Blood work (every year)	85 (22%)	58 (15%)	17 (4%)	20 (5%)	63 (16%)	243 (13%)
No regular blood work, only as needed	183 (48%)	222 (58%)	358 (92%)	319 (82%)	284 (73%)	1366 (71%)

COAST: Canine Osteoarthritis Staging Tool, CT: computerised tomography, MRI: Magnetic resonance imaging, OA: Osteoarthritis. ^1^ Percentages for each country are associated with local population.

**Table 5 animals-14-02231-t005:** Overview of patients’ previous treatments before initiation of bedinvetmab.

Previous treatments			
Yes		1199 (62%)	
No		733 (38%)	
Type of previous treatment			
Oral		1006 (84%)	
Physical Therapy		703 (59%)	
Injectable		229 (19%)	
Nutraceutical		137 (11%)	
Other		92 (8%)	
Mean number of previous treatments per patient			
Oral		1.6	
Injectable		1.0	
Steroids		1.1	
Other		1.0	
Oral treatments			
1		561 (48.3%)	
2		290 (27.9%)	
3		134 (21.4%)	
4		14 (1.2%)	
5		7 (1.2%)	
Previous oral treatments (*n* = 1199)			
Therapy	N (%)		Mean duration (days)
NSAIDs			
Metacam^®^	470 (46.7%)		58
Previcox^®^	326 (32.4%)		52
Galliprant^®^	267 (26.5%)		48
Onsior^®^	148 (14.7%)		52
Rimadyl^®^	75 (7.3%)		53
Trocoxil^®^	19 (1.5%)		55
Cimalgex^®^	10 (<1%)		46
Cimicoxib^®^	7 (<1%)		104
Non-NSAIDs			
Gabapentin	143 (14.2%)		64
Tramadol	100 (8.5%)		52
Amitryn^®^	19 (3%)		41
Dermipred^®^	45 (6%)		72
Amantadine	3 (<1%)		19
Paracetamol	2 (<1%)		1
Injectable treatments			
1		223 (97%)	
2		6 (3%)	
Previous injectable treatments (*n* = 229)			
Therapy	N (%)		Mean duration (days)
NSAIDs			
Rimadyl^®^	123 (52%)		26
Metacam^®^	70 (31%)		47
Onsior^®^	35 (15%)		4
Previcox^®^	1 (<1%)		35
Non-NSAIDs			
Tramadol	5 (2%)		20
Gabapentin	1 (<1%)		113
Nutraceuticals			
Any nutraceutical(s), of total PRFs		137 of 1932 (7%)	
1		123 of 137 (90%)	
2		13 of 137 (10%)	
3		1 of 137 (1%)	
Other interventions			
1		91 (99%)	
2		1 (1%)	
Previous other interventions (*n* = 93)			
Therapy		N (%)	
Hydrotherapy		43 (47%)	
Laser treatment		26 (28%)	
Cannabidiol oil		10 (11%)	
Gabapentin		6 (7%)	
Tramadol		5 (5%)	
JOINTSURE^®^		1 (1%)	
petMOD^®^		1 (1%)	
Physical therapy		1 (1%)	

**Table 6 animals-14-02231-t006:** Veterinarian satisfaction rating of bedinvetmab treatment.

Veterinarian Satisfaction Rating	Country ^1^						Severity ^1^		
	France (*n* = 381)	Germany (*n* = 384)	Italy (*n* = 390)	Spain (*n* = 387)	UK (*n* = 390)	Total (*n* = 1932)	Mild (*n* = 696)	Moderate (*n* = 712)	Severe (*n* = 524)
1	0.0%	0.0%	0.0%	0.0%	0.0%	0.0%	0.0%	0.0%	0.0%
2	0.0%	0.0%	0.0%	0.0%	0.0%	0.0%	0.0%	0.0%	0.0%
3	0.0%	0.0%	0.0%	0.8%	0.0%	0.2%	0.0%	0.3%	0.2%
4	0.8%	0.0%	0.8%	0.3%	0.0%	0.4%	0.0%	0.7%	0.4%
5	1.6%	0.0%	2.8%	0.5%	2.3%	1.4%	2.4%	1.3%	0.4%
6	1.8%	0.0%	7.4%	3.6%	8.7%	4.3%	5.5%	4.5%	2.7%
7	15.2%	12.2%	33.3%	18.1%	21.8%	20.2%	21.4%	19.8%	19.1%
8	42.3%	47.4%	39.2%	42.6%	39.2%	42.1%	40.1%	42.0%	45.0%
9	35.4%	36.2%	14.9%	31.0%	22.3%	27.9%	26.9%	27.9%	29.2%
10	2.9%	4.2%	1.5%	3.1%	5.6%	3.5%	3.7%	3.5%	3.1%
Mean score	8.1	8.3	7.6	8.1	7.9	8.0	7.9	8.0	8.1

^1^ N values refer to number of PRFs per subgroup.

**Table 7 animals-14-02231-t007:** Overall summary of use of other therapies after initiating bedinvetmab (further breakdowns are in Table 8, Table 9 and Table 10).

Therapy	Total (*n* = 1932)	Mild (*n* = 696)	Moderate (*n* = 712)	Severe (*n* = 524)
No other treatment since starting bedinvetmab	1449 (75.0%)	578 (83.1%)	524 (73.6%)	347 (66.2%)
Discontinuation of other therapies	22 (1.1%)	0 (0%)	9 (1.3%)	13 (2.5%)
Continuation of other therapies (more information is available in Table 8, Table 9 and Table 10)	189 (9.8%)	44 (6.3%)	75 (10.5%)	70 (13.4%)
Addition of other therapies	280 (14.5%)	76 (10.9%)	105 (14.8%)	99 (18.9%)

Note: Each column totals more than 100% due to multiple actions being taken after the initiation of bedinvetmab, in some cases.

**Table 8 animals-14-02231-t008:** Overview of discontinuation of other therapies after initiating bedinvetmab (subset of Table 7).

Discontinued Interventions					
Therapy	N (%)	Mild (*n* = 0)	Moderatev (*n* = 9)	Severe (*n* = 13)	Mean Duration (Days)
NSAIDs					
Metacam^®^	10 (45.5)	0	2	8	9
Previcox^®^	4 (18.2)	0	2	2	9
Galliprant^®^	2 (9.1)	0	0	2	12
Rimadyl^®^	1 (4.5)	0	1	0	8
Non-NSAIDs					
Gabapentin	3 (13.6)	0	2	1	7
Nutraceuticals	1 (4.5)	0	1	0	1
Tramadol	1 (4.5)	0	1	0	8

**Table 9 animals-14-02231-t009:** Overview of continuation of other therapies after initiating bedinvetmab (subset of Table 7).

Number of interventions continued (among those who continued)					
1					189 (98.4%)
2					3 (1.6%)
Continued interventions					
Therapy	N (%)	Mild (*n* = 44)	Moderate (*n* = 76)	Severe (*n* = 72)	Mean duration (days)
NSAIDs					
Metacam^®^	17 (9.0)	2	6	9	50
Previcox^®^	9 (4.8)	0	5	4	66
Galliprant^®^	7 (3.7)	0	4	3	51
Rimadyl^®^	3 (1.6)	1	2	0	23
Non-NSAIDs					
Physical therapy	70 (37.0)	26	26	18	79
Nutraceuticals	63 (33.3)	11	24	28	40
Tramadol	9 (4.8)	0	4	5	16
Hydrotherapy	6 (3.2)	2	2	2	20
Gabapentin	2 (1.1)	0	1	1	17
Laser therapy	2 (1.1)	1	0	1	13
Amitryn^®^	1 (0.5)	0	1	0	60
Cannabidiol oil	1 (0.5)	1	0	0	60

**Table 10 animals-14-02231-t010:** Overview of addition of other therapies after initiating bedinvetmab (subset of Table 7).

Number of interventions added (among those who added)					
1			280 (90.7%)		
2			26 (8.9%)		
3			1 (0.4%)		
Added interventions					
Therapy	N (%)	Mild (*n* = 86)		Moderate (*n* = 113)	Severe (*n* = 108)
NSAIDs					
Galliprant^®^	9 (3.2)	1		3	5
Metacam^®^	9 (3.2)	2		3	4
Previcox^®^	8 (2.9)	0		7	1
Rimadyl^®^	2 (0.7)	0		1	1
Onsior^®^	1 (0.4)	0		0	1
Non-NSAIDs					
Nutraceuticals	178 (63.6)	63		65	50
Tramadol	37 (13.2)	2		14	21
Physical therapy	20 (7.1)	6		8	6
Gabapentin	15 (5.4)	1		3	11
Cannabidiol oil	8 (2.9)	2		4	2
Amitryn^®^	3 (1.1)	2		0	1
Amantadine	1 (0.4)	0		1	0
Dermipred^®^	1 (0.4)	0		0	1
Hydrotherapy	1 (0.4)	1		0	0

**Table 11 animals-14-02231-t011:** Patients receiving multiple pharmacological therapies before and after initiation of bedinvetmab.

	All	UK	Germany	France	Italy	Spain	Mild	Moderate	Severe
n	1199	247	243	250	227	232	289	499	411
Before initiation	47%	43%	59%	53%	40%	38%	29%	45%	61%
After initiation	31% *	35% *	21% *	40% *	31% *	29% *	27%	30% *	36% *

* Significantly different to “Before initiation” (*p* < 0.05; assessed via paired *t*-tests at 95% confidence interval).

**Table 12 animals-14-02231-t012:** Average number of pharmacological medications received per patient, before and after initiation of bedinvetmab.

	All	UK	Germany	France	Italy	Spain	Mild	Moderate	Severe
n	1199	247	243	250	227	232	289	499	411
Before initiation	1.85	1.77	2.15	2.09	1.60	1.61	1.48	1.81	2.17
After initiation	1.33 *	1.36 *	1.25 *	1.43 *	1.32 *	1.30 *	1.28 *	1.32 *	1.39 *

* Significantly different to “Before initiation” (*p* < 0.05; assessed via paired *t*-tests at 95% confidence interval).

## Data Availability

Data are confidential, and consent was not gathered for sharing of data. However, fully anonymised results data are available upon reasonable request.

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
