# Peer review of "Use of Bedinvetmab (Librela®) for Canine Osteoarthritis in France, Germany, Italy, Spain, and the UK: Quantitative Analysis of Veterinarian Satisfaction and Real-World Treatment Patterns"

_animals, 2024, doi:10.3390/ani14152231_

Round 1
Reviewer 1 Report
Comments and Suggestions for Authors
Thank you for the opportunity to review the paper entitled:Use of bedinvetmab (Librela®) for canine osteoarthritis in France, Germany, Italy, Spain, and the UK: quantitative analysis of veterinarian satisfaction and real-world treatment patterns
Simple summary :
The simple summary and the abstract look the same.
I suggest the authors change the simple summary which should be written for a lay audience.
Line 12: OA it's an abbreviation, it would be better not to include it in the simple summary. Anyway, the full name should be indicated first.
Introduction:
Line 70: A brief explanation about trkA and p75NTR might be useful to the reader.
Material and Methods:
The inclusion criteria both regarding GP veterinary and the prescription times for bedinvetmab are not adequately explained. Based on what were these criteria chosen?
It should be also reported in detail how the GP veterinarian made the diagnosis (presumed or confirmed) of osteoartitis.
Furthermore, there is no description of why pregnant or lactating patients should be excluded from the patient selection criteria.
Results:
Table 4.
In the results section conflicting data are reported: the authors reported a mild, moderate and severe OA even if the 80% are reported not to have used a staging tool.
Conclusion:
I don't think the authors can conclude that the use of bedinvetmab is satisfactory in patients with OA.
In the materials and methods section the authors discussed about a suspected or confirmed diagnosis of OA.
Reference:
I recommend authors change reference numbers throughout the text, as indicated in the guidelines: “In the text, reference numbers should be placed in square brackets [ ], and placed before the punctuation; for example [1], [1–3] or [1,3].”
Reviewer 2 Report
Comments and Suggestions for Authors
Thank you for the opportunity to review this manuscript. This paper provides insight on early prescribing patterns of bedinvetmab in Europe following its initial launch. It demonstrated that bedinvetmab had improved compliance compared to other OA treatments and had a high level of satisfaction reported by the prescribing veterinarian as a sole-therapy or in combination with other treatments. Dogs receiving bedinvetmab were able to reduce the number of other OA treatments. Prescribing and use patterns were assessed in a varied population of dogs differing in size, age, and OA severity to provide insight into how these patterns might differ across different populations of dogs.
As this is a survey study on prescribing patterns, it does not provide information as to the efficacy and safety profile of bedinvetmab. The study was also conducted over a short period of time, therefore long-term or changes in prescribing patterns cannot be elucidated from the reported data. As the authors discuss, the data is also subject to several limitations such as self-selection and recall bias by the veterinary respondents.
Line 12: Please write out osteoarthritis here before using abbreviation of OA to ensure clarity for the reader.
Table 3: From this table it looks as though 100% of veterinarians surveyed were using bedinvetmab during the earliest period from Feb-April 2021. Since you describe that you required at least 50% of study respondents to be using bedinvetmab during this period and the reamining respondents could comprise veterinarians who began prescribing between May and September, I wanted to check to make sure that the data in the table is accurate and that all veterinarians surveyed were indeed prescribing bedinvetmab during the earliest months of availability.
Table 4: Any thoughts on why more male dogs represented vs female?
Table 5: This is a very long table. Would it be possible to group nutraceuticals into categories of the main component, such as glucosamine/chondroitin, omega-3s, etc. Also, there are several items listed under both ‘previous nutraceuticals’ as well as ‘previous other interventions’ including Jointsure and petMOD, please edit to avoid repetition.
Reviewer 3 Report
Comments and Suggestions for Authors
I have very much enjoyed reading this manuscript and have very few, minor comments.
Introduction
Line 52: ‘showed’ might be better written as, ‘shown’.
Line 61: ‘liver’ might be better as, ‘hepatic’.
Line 62: ‘Efficacy with NSAID use improves over time’. I think the authors mean ‘NSAID efficacy improves with their use over time’.
Materials and Methods
Line 153: I think that ‘Table 1’ should be ‘Table 2’.
Lines 192-3: I am intrigued that outliers were removed. How many outliers were identified and how sure were the authors that these were not worthy of inclusion?
Lines 202-205: Did the authors check for normality of data distributions before applying parametric statistical analyses?
Results
Lines 213-4: I see an error message, but I believe this should be a reference to Table 3.
Lines 205-6: I see an error message, but I think this should be a reference to Table 4.
Page 6 of 21; Table 3
Where Rimadyl is first mentioned in the left hand column, I believe there should be a ‘registered trade mark’ sign next to it.
If only one brand of each NSAID was reported, I wondered if it would be easier to use generic names?
What drugs were included in the category of ‘Generic NSAID’?
Lines 231, 235, 243 and 255: I see error messages, but I believe these are references to Table 5.
Page 10 of 21; Table 5
Should there be a ‘registered trade mark’ sign following: Jointsure, NUTRICH, Maxxiflex+, ARTRO, Condrogen energy and Prometheus hip and joint tablets?
I think that MSM should be written as, ‘MSM (methylsulfonylmethane)’.
Lines 271, 274 and 276: I see error messages but think these are references to Table 4.
Line 279: I see an error message but I think this should be a reference to Figure 2.
Lines 293-294: I see an error message but I think this should be a reference to Table 6.
Line 323: ‘a mean rating 2.5 points higher’, I think should be, ‘a mean rating 1.5 points higher’.
Page 15 of 21; Table 9
I think ‘Amitryn’ should be followed by the symbol for ‘registered trade mark’.
Round 2
Reviewer 1 Report
Comments and Suggestions for Authors
Dear Authors,
Thank you for revising your paper.
The following sentence appears several times throughout the text, please check and correct: "see Error! Reference source not found".
Line 221, 223, 239, 243, 249, 262, 277, 281, 283, 286, 300
Author Response
Dear reviewer,
Thank you for raising this issue with the cross-references within the document.
We have now converted these cross-references to plain text, to avoid this issue.
Best wishes,
The author team